# Multi-Omics Unveils Inflammatory Regulation of Fermented Sini Decoction Dregs in Broilers Infected with Avian Pathogenic *Escherichia coli*

**DOI:** 10.3390/vetsci12050479

**Published:** 2025-05-15

**Authors:** Shuanghao Mo, Xin Fang, Wenxi Xiao, Bowen Huang, Chunsheng Li, Hui Yang, Yilin Wu, Yiming Wang, Hongxia Ma

**Affiliations:** 1College of Animal Science and Technology, College of Veterinary Medicine, Jilin Agricultural University, Changchun 130118, China; mosh0907@mails.jlau.edu.cn (S.M.); 18043281905@163.com (X.F.); huangbw1999@163.com (B.H.); jluyanghui163@163.com (H.Y.); ealin_1088@163.com (Y.W.); hongxia0731001@163.com (H.M.); 2Jilin Provincial Key Laboratory of New Veterinary Drug R&D and Creation, Changchun 130118, China; 3College of Life Sciences, Jilin Agricultural University, Changchun 130118, China; 4The Engineering Research Center of Bioreactor and Drug Development, Ministry of Education, Jilin Agricultural University, Xincheng Street No. 2888, Changchun 130118, China

**Keywords:** avian pathogenic *Escherichia coli* (APEC), fermented liquid of sini decoction dreg (FLSDD), anti-inflammatory

## Abstract

Avian colibacillosis, a common bacterial disease in poultry caused by avian pathogenic *Escherichia coli*, leads to significant economic losses in the agricultural industry and poses risks to human health through potential food safety issues. This study addressed the growing problem of antibiotic resistance, which complicates disease control, by seeking alternative strategies. Our goal was to develop a safe, effective treatment using fermented dregs from traditional Chinese medicine, processed with *Lactobacillus rhamnosus* strain, to combat the infection. Through analyzing the transcriptome data of the *GEO* database, we identified 11 key genes, including *TLR4*, which play a vital role in recognizing and fighting the bacteria by activating immune defenses. We also tested natural compounds, *Dioscin* and *Celastrol*, extracted from the fermented liquid, finding they reduced inflammation triggered by the infection. The fermented liquid itself proved more effective in live animals, likely due to the combined action of multiple beneficial components. This approach offers a sustainable method to manage avian colibacillosis, decrease reliance on antibiotics, and enhance poultry health. These advancements could support efforts to reduce the cost of poultry farming, improve animal welfare, and ensure society produces safer food.

## 1. Introduction

Traditional Chinese medicine (TCM), developed over thousands of years, with a long history and remarkable pharmacological effects, has been extensively studied in the field of pharmacology and widely applied in various therapeutic approaches until now. TCM is generally categorized into single-herb and multi-herb formulas. Compared to single-herb formulations, multi-herb formulas exhibit higher efficacy and versatility in disease prevention and treatment [1,2].

The primary resources of TCM include three categories: herbal medicines, animal-derived medicines, and mineral medicines. Among these, herbal medicines account for 87% of the resources. However, in the production of TCM products, only an average of 5% of the active ingredients are extracted and utilized. Systematic studies on *Panax notoginseng* residues have shown that traditional TCM companies primarily extract single active ingredients, resulting in the discard of large amounts of organic nutrients, such as cellulose, hemicellulose, lignin, proteins, nucleic acids, and trace elements [3]. Moreover, some effective components are not fully extracted, leading to significant resource waste [4]. According to incomplete statistics, TCM manufacturers consume millions of tons of herbal materials annually, generating tens of millions of tons of medicinal wastewater and dregs [5].

To address this issue, researchers have implemented various effective strategies and methods to make full use of the medicinal dregs. These measures include using dregs as a substrate for edible mushrooms, converting dregs into organic fertilizers, and utilizing dregs as feed additives and fuel [6].

Avian colibacillosis, caused by specific serotypes of opportunistic pathogenic *Escherichia coli* [7], can occur throughout the year and affect chickens of all ages, to which chicks and broilers are particularly susceptible. This disease presents a range of typical symptoms, including acute septicemia, salpingitis, peritonitis, panophthalmitis, arthritis, omphalitis, coligranuloma, and early embryo mortality. The main routes of infection are the respiratory and digestive tracts [8]. Additionally, vertical transmission through eggs and feces-contaminated eggs is a key pathway for disease spread [9].

Probiotics are a group of active microorganisms that are beneficial to the host. They are introduced into the gastrointestinal and reproductive systems of humans and animals, where they produce active compounds proven to benefit health. These benefits include enhancing the hosts’ microecological balance and promoting beneficial gut functions. Globally, research on probiotics has become a focus due to their ability not only to promote animal growth, regulate normal gut microbiome, maintain ecological balance and improve gastrointestinal function, but also enhance food digestibility and bioavailability, reduce serum cholesterol, control endotoxins, inhibit the growth of harmful bacteria in the gut, boost immunity and help prevent and treat various conditions including gastrointestinal syndromes and respiratory infections [10,11]. Among many probiotic strains, *Lactobacillus rhamnosus* is the best-known and most extensively studied one. It can adapt to the digestive environment, utilize prebiotics to stimulate growth, and play a key role in balancing and enhancing gut function, promoting the growth and activity of *Bifidobacterium* and *Lactobacillus acidophilus*, preventing diarrhea and reducing respiratory infections [12].

This study conducted an in-depth investigation and analysis of transcriptome data *GSE69014* from the Gene Expression Omnibus (*GEO*) database and utilized *Lactobacillus* derived from food sources to ferment the Sini decoction dregs (SDD), which were then fed to broilers. Additionally, Liquid Chromatography–Mass Spectrometry (LC–MS) was employed to identify the components of the fermented liquid of Sini decoction dregs (FLSDD), followed by bioinformatics analysis to determine its pharmacological activity. This approach aims to elucidate the key molecular mechanisms of avian pathogenic *Escherichia coli* infectious inflammation in broilers and explore potential feasible prevention and treatment strategies.

## 2. Materials and Methods

### 2.1. Transcriptome Sequencing Datasets

The transcriptome sequencing dataset *GSE69014* [13] was obtained from the *GEO* database. In the *GSE69014* dataset, male broilers were challenged with avian pathogenic *Escherichia coli* (APEC), and the thymuses were collected on days 1 and 5 post-infection. Based on autopsy scores of the liver, pericardium, and air sacs, the infected chickens were classified into mild and severe categories, representing resistant (R) and susceptible (S) phenotypes, respectively. Uninfected chickens were categorized as the healthy group (NC). All gene expression data measurements are available for download from the *GEO* database.

### 2.2. Differential Expression Genes (DEGs) Analysis

The dataset was read using R Studio (R 4.3.2) and preprocessed using the dplyr package. Subsequently, the ggplot2, tidyverse, and ggrepel packages were used to perform gene expression significance analysis and DEGs analysis on the *GSE69014* dataset, and a volcano plot was generated.

### 2.3. Weighted Gene Co-Expression Network Analysis (WGCNA)

Weighted Gene Co-expression Network Analysis (WGCNA) constructs a co-expression network from gene expression data and uses hierarchical clustering to group genes into functionally related modules, thereby revealing interactions between genes and their impact on biological traits. Using the WGCNA R package, highly co-expressed gene groups were identified in the *GSE69014* datasets, and genes were classified into different modules based on their expression patterns to elucidate potential biological pathways and pathogenic mechanisms of APEC infection in chickens. Furthermore, genes from modules highly correlated with external traits were evaluated and selected as candidate genes associated with WGCNA analysis results.

### 2.4. Protein–Protein Interaction (PPI) Network

For the key genes identified in the *GSE69014* dataset, the *STRING* database was used to predict and visualize the protein–protein interaction (PPI) network. The resulting network was then imported into Cytoscape v3.9.1, and the CytoHubba plugin was used to identify a set of hub genes as biomarkers for APEC infection.

### 2.5. CIBERSORT Immune Infiltration Analysis

In order to evaluate the composition and relative abundance of immune cells in each sample of *GSE69014*, we used the CIBERSORT (Cell-type Identification By Estimating Relative Subsets Of RNA Transcripts) algorithm to analyze the transcriptome data. CIBERSORT is a computational method based on support vector regression (SVR) that can use gene expression data to infer the relative proportions of 22 immune cell subsets in a sample. In order to explore the potential association between gene expression levels and immune cell infiltration, we used R to calculate the Spearman correlation coefficient between the expression levels of each gene and the proportion of immune cells inferred by CIBERSORT and used ggplot2 to draw stacked bar charts and correlation heat maps.

### 2.6. Gene Enrichment Analysis

Samples from the *GSE69014* dataset were divided into a healthy group and an APEC-infected group (susceptible(S) phenotypes). GSEA enrichment analysis was performed using R packages such as clusterProfiler and DESeq2, and key pathways were identified by examining the enrichment results. Key pathways related to APEC infection in broilers were identified by integrating bioinformatics databases (KEGG) and literature research.

### 2.7. Preparation of Sini Decoction Dregs Fermentation Liquid

Sini decoction (SD) is a compound traditional Chinese medicine formula composed of *Angelica sinensis*, *jujube*, *cinnamon*, *red peony root*, *Aristolochia debilis*, *licorice*, and *Asarum*. All medicinal herbs were provided by Beijing Ben Cao Fang Yuan (Bozhou) Pharmaceutical Technology Co., Ltd. (Bozhou, China), and were subjected to two rounds of water decoction to prepare the SDD before fermentation. The *Lactobacillus rhamnosus* strain used was isolated from kimchi and preserved at the China Center for Type Culture Collection (CCTCC), with strain number CCTCC M 2022083. First, 30 g of SDD, which had been subjected to two rounds of water decoction and dried, were mixed with 300 mL of distilled water and sterilized at 121 °C for 20 min, then cooled to room temperature to obtain the unfermented SDD dregs extract. Subsequently, under aseptic conditions, 1 mL of overnight-cultured (12 h) *L. rhamnosus* inoculum was added to the cooled dregs aqueous extract (AE) and incubated at 37 °C and 160 rpm for 12 h to produce the fermented liquid. After fermentation, the mixture was centrifuged at 10,000 rpm for 10 min to obtain the supernatant as the FLSDD, which was stored at −20 °C until use.

### 2.8. Metabolites Extraction

FLSDD samples were thawed at 4 °C and vortexed for 1 min. Then, an appropriate amount of the sample was accurately transferred to a 2 mL centrifuge tube, followed by the addition of 400 µL methanol, and vortexed for another minute. The mixture was then centrifuged at 12,000 rpm for 10 min at 4 °C, and the supernatant was transferred to a new 2 mL centrifuge tube and concentrated to dryness. The dried sample was reconstituted with 150 µL of 80% methanol solution containing 2-chloro-L-phenylalanine (4 ppm), and the reconstituted sample was filtered through a 0.22 μm membrane filter. The filtrate was collected into a vial for LC–MS analysis.

### 2.9. LC–MS/MS Analysis

Chromatographic conditions: A Thermo Scientific™ Vanquish™ Horizon UHPLC system (Thermo Fisher Scientific, Germering, Germany; address: Sommerstrasse 6, 82110 Germering, Germany; www.thermofisher.com, accessed on 13 May 2025) was used, equipped with an ACQUITY UPLC^®^ HSS T3 column (2.1 × 100 mm, 1.8 µm) (Waters Corporation, Milford, MA, USA; address: 34 Maple Street, Milford, MA 01757, USA; www.waters.com, accessed on 13 May 2025). The column temperature was maintained at 40 °C, and the flow rate was set to 0.3 mL/min. The mobile phase consisted of acetonitrile with 0.1% formic acid and water. The gradient elution started with 8% acetonitrile, increased to 98% over 8 min, maintained at 98% until the 10th minute, then quickly dropped back to 8%, and continued until the 12th minute.

Mass spectrometry conditions: A Thermo Q Exactive Focus mass spectrometer was used, equipped with an electrospray ionization source. Data were collected in both positive and negative ion modes. The capillary temperature was set to 325 °C. In full scan mode, the m/z range was 100 to 1000, with a resolution of 70,000 and a collision energy of 30 eV.

### 2.10. Molecular Docking

The crystal structures of protein (*3VQ2*) were downloaded from the PDB database (https://www.rcsb.org/, accessed on 13 May 2025) and processed using AutoDockTools 1.5.6, which included removing ligands and water, adding hydrogen atoms, and saving the files in pdbqt format to be used as receptors for molecular docking. The selected ligand compounds (potential active components from FLSDD) were hydrogenated and converted into 3D coordinates using RDKit, and then converted to pdbqt format using Open Babel 2.4.1 to serve as ligands for molecular docking. Molecular docking was performed using AutoDock Vina, and the docking results were visualized using Pymol 2.5.7 and Discovery Studio 2024 Client.

### 2.11. Construction of Cell Inflammation Model

Chicken macrophages *HD11* were inoculated in 6-well plates at 1 × 10^6^/well and cultured in a 37 °C incubator for 12 h. The control group received no treatment. AE, FLSDD, *Dioscin* (DIO) and *Celastrol* (CEL) were added to each experimental group. Except for the control group, the rest of the experimental groups were inoculated with APEC in the well plate at a multiplicity of infection MOI = 0.1 and cultured at 37 °C for 6 h. The specific treatment methods are shown in Table 1.

### 2.12. Construction of Broiler Prevention Model

A total of 105 3-day-old Ross broilers were randomly divided into seven groups. The control group received no treatment. AE, FLSDD, DIO, and CEL were added to each experimental group. After 7 days of feeding, all treatment groups except the control group were challenged orally. The mental state of the birds was observed every 8 h, and after 24 h, blood samples were collected, and the birds were euthanized by cervical dislocation, followed by the collection of the duodenum. The detailed treatment methods are shown in Table 2.

### 2.13. Total Protein Extraction and Western Blot

All experimental antibodies were purchased from Wuhan Proteintech Co., Ltd. (Proteintech Group, Wuhan, Hubei, China; address: 666 Gaoxin Avenue, East Lake High-tech Development Zone, Wuhan 430075, Hubei, China; www.ptgcn.com, accessed on 13 May 2025). Chicken macrophages *HD11* were washed with pre-chilled PBS, and 400 μL of RIPA lysis buffer was added. After lysis on ice for 10 min, the mixture was transferred to a 1.5 mL centrifuge tube and lysed on ice for an additional 30 min. It was then centrifuged at 4 °C and 12,000 r/min for 15 min, and the supernatant was collected. Chicken duodenal tissue was quick-frozen in liquid nitrogen, weighed, and thoroughly ground. For every 100 mg of solid tissue, 1 mL of pre-chilled lysis buffer (RIPA lysis buffer + PMSF) was added. The mixture was lysed on ice for 30 min, and the suspension was centrifuged at 12,000 r/min and 4 °C for 15 min, after which the supernatant was collected. The protein concentrations of all samples were determined using a BCA protein quantification kit and normalized. Based on the size of the target protein, SDS-PAGE gels were prepared. Protein samples and markers were sequentially loaded into the wells, electrophoresed at 80 V for 20 min, then switched to 120 V until the loading buffer migrated to the bottom. The proteins were transferred to a PVDF membrane at 200 mA for 60 min and blocked with 5% skim milk powder for 2 h. Primary antibodies were incubated at 4 °C for 12 h, followed by secondary antibodies for 2 h. Excess antibodies were washed off with TBST, and ECL luminescent solution was used for development in a protein imaging system, *β-actin* was used as an internal reference, and ImageJ (ImageJ bundled with 64-bit Java 8) was employed for grayscale analysis to quantify the expression levels of the target proteins.

### 2.14. Enzyme-Linked Immunosorbent Assay (ELISA)

The ELISA kit was purchased from Shanghai Enzyme-linked Biotechnology Co., Ltd. (Shanghai, China; address: 5500 Yuanjiang Road, Minhang District, Shanghai 201111, China; en.mlbio.cn, accessed on 13 May 2025). Blood samples were collected from the wing vein of broilers in each group to obtain serum, and the levels of *IL-1β*, *IL-4*, *IL-6*, *IL-8*, *IL-10*, *IL-12*, *TNF-α*, and *iNOS* were measured according to the manufacturer’s instructions.

## 3. Results

### 3.1. Transcriptome Analysis Results

#### 3.1.1. Identification of DEGs

After processing the transcriptome sequencing dataset GSE69014, genes with |Log_2_FC| ≥ 0.6 and *p* ≤ 0.05 were selected as DEGs. As shown in Figure 1, A total of 1034 DEGs were identified in the GSE69014 dataset (943 upregulated, 91 downregulated). Compared to the resistant group (R), the susceptible phenotype (S) broilers showed more DEGs post-infection.

#### 3.1.2. WGCNA and Identification of Key Modules

Data cleaning was performed on the 24 transcriptome samples from GSE69014: first, clustering analysis was conducted to exclude outlier samples (height ≥ 100,000) and genes with low expression and poor quality (fragments per kilobase of transcript per million mapped reads ≤ 100), Figure 2a. After identifying abnormal values or outliers, hierarchical clustering was conducted on the remaining genes in GSE69014. Based on the scale independence (Figure 2b) and mean connectivity (Figure 2c) of the network, a soft threshold of 20 was selected to ensure a scale-free topology. Dynamic tree cutting was used to identify gene modules, and dendrograms and module colors were visualized, merging highly correlated modules; see Figure 2d. By correlating with external traits, the relationship between module eigengenes and external sample traits was calculated, and a heatmap was plotted. As shown in Figure 2e, two gene modules related to the susceptible phenotype (S) were identified in GSE69014: *MEblack* and *MEbrown*. The 2129 genes in these modules were considered potential genes associated with APEC infection.

#### 3.1.3. Construction of PPI Network

A comprehensive analysis of the DEGs and WGCNA results from dataset GSE69014 identified 665 key genes associated with severe APEC infection in broilers. A PPI network was constructed using *STRING* and imported into Cytoscape, where *TLR4* was found at the core of the PPI network, suggesting its significant biological importance; see Figure 3.

#### 3.1.4. GSEA Results

The GSEA enrichment analysis of dataset GSE69014 revealed that 23 pathways were significantly upregulated and 25 pathways were significantly downregulated (|Normalized Enrichment Score, NES| ≥ 1.5, *p* ≤ 0.05), Figure 4a. In addition, single-gene enrichment analysis of *TLR4* revealed that among the six signaling pathways it is involved in, two were significantly activated. These are the Phagosome signaling pathway (NES = 1.86, *p* < 0.0001) and the Toll-like receptor signaling pathway (NES = 1.57, *p* < 0.02), Figure 4b.

#### 3.1.5. CIBERSORT Immune Infiltration Results

The immune infiltration profiles of 24 thymic samples from the GSE69014 dataset were simulated using the CIBERSORT algorithm. The proportions of immune cells in each sample and their intergroup differences are presented in Figure 5a,c. Although the differences between groups for most immune cells were not statistically significant, correlation analysis between these cell proportions and core gene expression levels revealed significant associations for certain immune cell types. Notably, apart from thymic cells, *M1* and *M2 macrophage* subtypes exhibited significant positive correlations with most core genes, with *TLR4* showing a Pearson correlation coefficient of 0.52 (*p* < 0.01) with both subtypes, Figure 5b,d.

#### 3.1.6. Components of FLSDD

LC–MS identified a total of 7527 natural compounds and metabolites in the FLSDD. By querying *DrugBank*, *TCMSP*, and *HERB* databases, a total of eight compounds targeting *TLR4* were obtained; see Figure 6 and Table 3.

#### 3.1.7. Molecular Docking Results

The molecular docking results of *TLR4* and eight components in the FLSDD are shown in Figure 7a. Among them, two compounds (*Dioscin* and *Celastrol*) exhibited high affinity for the *TLR4*-MD2 complex, with binding energies of −10.8 kcal/mol and −10.3 kcal/mol, respectively. Both compounds were able to occupy the hydrophobic pocket in *MD2* that binds to lipid A of lipopolysaccharide (LPS); see Figure 7b.

### 3.2. Cell and Animal Experiments Results

#### 3.2.1. Western Blot

Western blot analysis was performed to assess key proteins in the *Toll-like receptor signaling pathway* and its downstream *NF-κB* and *MAPK* cascades in chicken macrophages *HD11*(Figure 8a) and duodenum(Figure 8b) across treatment groups. Results revealed that in the APEC infection group (IG), *TLR4* and *MyD88* levels were markedly elevated compared to the control group (NC) (*p* ≤ 0.01). APEC infection also significantly increased the phosphorylation of *p38*, *JUN*, *IκBα*, and *p65*, with a consistent rise observed at both cellular and organismal levels. In cell models, *Dioscin* (DIO) and *Celastrol* (CEL) treatments showed slightly better inhibition of these proteins’ elevated expression and phosphorylation compared to FLSDD and dregs aqueous extract (AE) groups. However, in animal models, their preventive effects were generally outperformed by the FLSDD treatment group.

#### 3.2.2. ELISA

The ELISA quantitative detection results of cytokines showed that, compared to the NC, all detected cytokine levels in the IG were significantly elevated, indicating that APEC infection induced a strong inflammatory response. This inflammatory response was effectively suppressed by AE. FLSDD, DIO, and CEL, as representative active components in FLSDD, showed some inhibitory effects on certain inflammatory cytokines, but their effects were not as strong as FLSDD; see Figure 9.

## 4. Discussion

Avian colibacillosis is one of the most common bacterial diseases in livestock and poultry, causing substantial economic losses to the farming industry and posing potential threats to human health and food safety [25]. With the expansion of farming operations and environmental impacts, the incidence of avian colibacillosis has been increasing [26]. Current treatment mainly relies on antimicrobial drugs, but long-term misuse has led to an increase in resistant strains, making disease prevention and control more challenging [27]. Therefore, there is an urgent need to find new antibacterial agents and anti-infection strategies to address resistance issues and effectively control the disease. According to incomplete statistics, TCM manufacturers consume millions of tons of plant medicinal materials annually, generating tens of millions of tons of medicinal dregs [28]. TCM has various pharmacological activities, and fermentation of medicinal dregs using probiotics such as *Lactobacillus* can achieve comprehensive utilization of TCM waste while enhancing its bioactivity [29]. This process combines the advantages of TCM, enzymes, probiotics, and secondary metabolites, providing a safe and efficient method that facilitates animal absorption. It holds significant practical implications for preventing and controlling livestock and poultry diseases like avian colibacillosis, reducing resistance, and improving production performance [30].

By mining the transcriptomic data of 24 chicken thymus samples from GSE69014, we identified 11 core genes, including *TLR4* (*CD36*, *C1QA*, *CSF1R*, *TLR4*, *CD80*, *STAT3*, *MMP9*, *CD40*, *IL15*, *CTSS*, *H6PD*), which collectively regulate immune and inflammatory responses during APEC infection. Among them, *CD36* and *TLR4*, as pattern recognition receptors, recognize bacterial LPS to initiate innate immune responses [31,32]; *C1QA* promotes bacterial clearance by activating the complement system [33]; *CSF1R* regulates macrophage differentiation and function, enhancing phagocytosis [34]; *CD80* and *CD40* play critical roles in antigen presentation and T-cell activation, facilitating adaptive immune responses [35]; *STAT3* and *IL15* maintain immune balance by regulating cytokine signaling and immune cell proliferation [36]; *MMP9* promotes immune cell migration by degrading the extracellular matrix [37]; *CTSS* participates in antigen processing and MHC-II molecule presentation [38]; and *H6PD* may indirectly influence immune responses by modulating oxidative stress [39]. Together, these genes orchestrate the immune defense mechanisms against APEC infection.

The *TLR4* gene encodes Toll-like receptor 4, a critical pattern recognition receptor that recognizes LPS from Gram-negative bacteria [40], playing a central role in phagosome signaling and Toll-like receptor signaling pathways. GSEA enrichment analysis revealed that in the susceptible (S) phenotype, the Toll-like receptor signaling pathway involving *TLR4*, along with *NF-κB* and *MAPK* cascades, exhibited significant activation. Located on the phagosome membrane, *TLR4* senses LPS and activates the *Toll-like receptor signaling pathway* via the adaptor protein *MyD88*, while also initiating signals within the phagosome to promote bacterial phagocytosis and degradation [41]. Downstream of the *Toll-like receptor signaling pathway*, *TLR4* activates *NF-κB* and *MAPK* cascades: *NF-κB* induces the expression of pro-inflammatory cytokines (e.g., *TNF-α*, *IL-6*) through *IκB* phosphorylation and nuclear translocation, thereby enhancing the inflammatory response; the *MAPK* pathway (e.g., *p38*) regulates the activity of the *AP-1* (*JUN*) transcription factor, further promoting the expression of inflammation- and immune-related genes [42]. These cascades collectively amplify immune signals, recruiting and activating immune cells such as macrophages and neutrophils to effectively clear bacterial infections. Additionally, immune infiltration analysis showed a positive correlation between *TLR4* expression levels and the proportions of M1 and M2 macrophage subtypes, suggesting that *TLR4* may play a significant role in macrophage polarization: by recognizing LPS from Gram-negative bacteria, *TLR4* activates signaling pathways that promote the polarization of pro-inflammatory M1 macrophages to enhance antibacterial phagocytosis and inflammation [43], while potentially inducing M2 macrophage generation through negative feedback or crosstalk signals, thereby facilitating tissue repair and inflammation resolution [44].

*MD2* is a small glycoprotein closely associated with *TLR4*, featuring a hydrophobic pocket on its surface specifically designed to accommodate the fatty acid chains of lipid A in LPS [45]. In the *3VQ2* structure, the hydrophobic fatty acid chains of lipid A insert into *MD2*’s hydrophobic pocket through van der Waals forces and hydrophobic interactions, forming tight Alkyl and Pi–Alkyl interactions with hydrophobic amino acids such as PHE (C:119, C:121, C:151, etc.), LEU (C:54, C:94, etc.), ILE (C:52, C:124, etc.), and VAL (C:63, C:61, etc.). Meanwhile, the phosphate groups of lipid A interact with the positively charged LYS (A:263) via ionic interactions and salt bridges, enhancing binding stability, while its hydroxyl and oxygen atoms form hydrogen bonds with SER (C:120) and GLU (C:122), further stabilizing the complex. This binding mode induces a conformational change in *MD2*, promoting *TLR4* dimerization and thereby initiating downstream immune signaling [46]. Among the natural products identified in FLSDD, DIO possesses multiple cyclic structures and hydrophobic side chains, resembling the hydrophobic fatty acid chains of lipid A. Molecular docking results show that DIO competitively inserts into *MD2*’s hydrophobic pocket, forming Alkyl or Pi–Alkyl interactions with PHE (C:119, C:121), LEU (C:87), and others, while its hydroxyl and polar groups form hydrogen bonds with SER (B:413, B:439), GLU (B:437), and others, occupying key binding sites on *MD2*. This suggests that DIO may interfere with LPS binding to the *TLR4-MD2* complex through competitive binding or steric hindrance. Similarly, CEL, which also contains multiple cyclic frameworks and hydrophobic side chains, can insert into *MD2*’s hydrophobic pocket, forming Alkyl and Pi–Alkyl interactions with PHE (C:151), ILE (C:52, C:46), and others, while its hydroxyl or polar groups form hydrogen bonds with SER (C:48) and VAL (C:61), occupying critical binding sites on *MD2*, potentially disrupting the normal binding of LPS to the *TLR4-MD2* complex.

Western blot analysis revealed that in chicken macrophages *HD11* and duodenum, the IG exhibited significantly elevated expression levels of *TLR4* and *MyD88* compared to the NC (*p* ≤ 0.01), alongside a marked increase in the phosphorylation levels of *p38*, *JUN*, *IκBα*, and *p65*. This trend was consistent at both cellular and organismal levels, indicating that APEC infection strongly induces a pro-inflammatory response by activating the *TLR4*-*MyD88* signaling pathway and its downstream *NF-κB* and *MAPK* cascades. Combined with ELISA results, all detected cytokines (e.g., *TNF-α*, *IL-1β*, *IL-6*, *IL-8*, *IL-12*) in the IG were significantly higher than in the NC, further confirming that APEC infection triggers a robust systemic inflammatory response. In cellular models, the DIO and CEL treatment groups showed slightly better inhibition of the expression and phosphorylation of proteins in the *Toll-like receptor signaling pathway* compared to the FL and AE, suggesting that DIO and CEL may interfere with LPS binding to the *TLR4-MD2* complex by competitively occupying *MD2*’s hydrophobic pocket, thereby suppressing pathway activation. However, in animal models, the FL demonstrated an overall superior preventive effect compared to the DIO and CEL groups, with ELISA results also showing that FLSDD more effectively suppressed *TNF-α*, *IL-1β*, *IL-6*, *IL-8*, and *IL-12* levels. This may indicate that other components in FLSDD, beyond DIO and CEL, exert synergistic effects in vivo, enhancing its anti-inflammatory activity; alternatively, FLSDD might indirectly suppress *TLR4* signaling pathway activation by modulating the gut microbiota and its metabolites, thus more effectively reducing systemic inflammation. The weaker performance of DIO and CEL in animal models compared to in vitro cell experiments may be attributed to their metabolic stability or bioavailability in vivo [47,48], potentially resulting in insufficient effective concentrations to sustain inflammation suppression. The anti-inflammatory efficacy of FLSDD may be linked to its enrichment with other potent anti-inflammatory components, though virtual drug screening based on molecular docking did not account for drug metabolism or absorption rates [49], and the specific components require further isolation and identification through additional methods.

While FLSDD demonstrates promising anti-inflammatory effects against APEC infection, its application as a natural product faces challenges related to component variability and standardization. The complex composition of FLSDD, influenced by factors such as raw material sources and fermentation conditions, may lead to batch-to-batch variations in the content and proportion of active compounds, potentially affecting therapeutic consistency [50]. Furthermore, the synergistic effects among multiple components, while contributing to FLSDD’s efficacy, are difficult to systematically characterize and track, complicating quality control. The observed weaker in vivo performance of individual compounds like DIO and CEL compared to in vitro results suggests potential limitations in their metabolic stability and bioavailability, which may also apply to other FLSDD components.

To address these challenges, future research should focus on establishing robust quality control standards, such as quantitative analysis of key active compounds using LC–MS and chemical fingerprinting to ensure batch consistency [51]. Standardized production processes, including controlled raw material sourcing and fermentation conditions, are essential to minimize variability. Additionally, network pharmacology and metabolomics approaches can elucidate the mechanisms of synergistic effects [52], while pharmacokinetic studies will optimize the bioavailability of active components. These efforts will enhance the reliability and scalability of FLSDD as a novel anti-inflammatory agent, supporting its potential in combating avian colibacillosis and promoting sustainable poultry farming.

## 5. Conclusions

This study systematically elucidated the critical role of *TLR4* in APEC infection and its regulatory mechanisms through transcriptomic analysis, molecular docking, Western blot, and ELISA experiments, while also validating the potential of FLSDD and its active components, DIO and CEL, in suppressing inflammatory responses. FLSDD, through the synergistic action of multiple components, demonstrated superior anti-inflammatory effects in animal models compared to individual compounds, providing a scientific basis for developing novel antibacterial and anti-infection strategies using traditional Chinese medicine dregs. This not only offers new insights into addressing antimicrobial resistance in avian colibacillosis but also paves the way for the efficient utilization of Chinese medicinal resources and the prevention and control of livestock and poultry diseases. Future research can focus on further isolating and identifying other active components in FLSDD, combined with pharmacokinetic studies, to optimize its application strategies in preventing and treating avian colibacillosis, thereby better supporting the sustainable development of the farming industry, reducing economic losses, and ensuring food safety.

## Figures and Tables

**Figure 1 vetsci-12-00479-f001:**
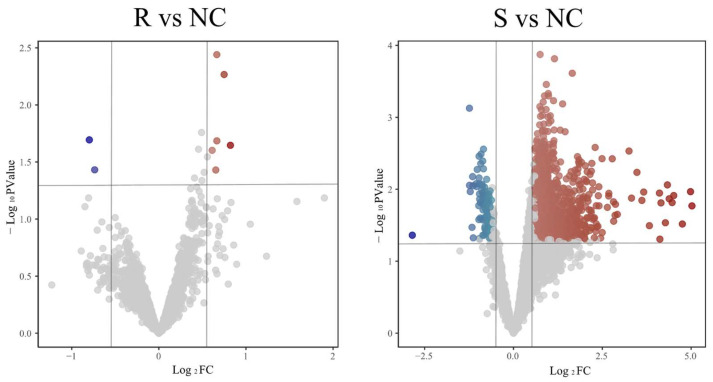
Volcano plot of DEGs group comparison. R represents the resistant phenotype, S represents the susceptible phenotype, and the healthy group is denoted by NC. Log_2_ Fold Change (Log_2_FC) represents the logarithm to base 2 of the ratio of gene expression levels under two conditions, indicating the magnitude and direction of differential gene expression. Red dots represent significantly up-regulated genes, and blue dots represent significantly down-regulated genes (|Log_2_FC| ≥ 0.6 and *p* ≤ 0.05). Gray dots represent non-significantly changed genes (|Log_2_FC| < 0.6 or *p* > 0.05).

**Figure 2 vetsci-12-00479-f002:**
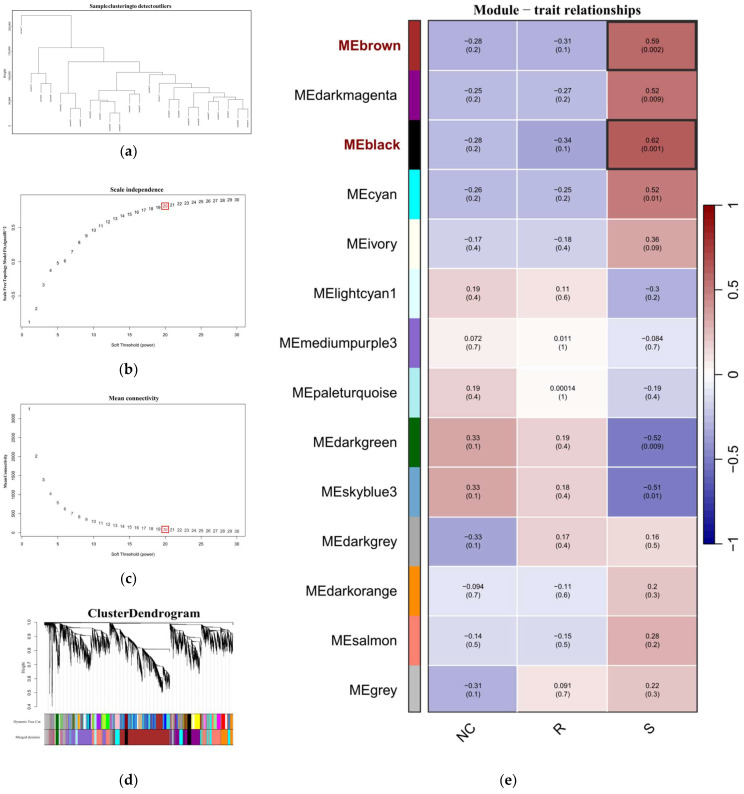
(**a**) The sample dendrogram, showing the similarity between samples through the dendrogram; (**b**,**c**) the soft-thresholding selection plot, which illustrates the network’s topological properties under different soft-thresholding powers (soft threshold power, defined as the exponent β applied to the correlation matrix to enhance strong gene correlations and construct a scale-free network). (**b**) shows the scale independence, indicating how well the network fits a scale-free topology (R^2^), while (**c**) displays the mean connectivity index of the scale-free topology model. By selecting an appropriate soft-thresholding power (β = 20 in this analysis), the network can maintain scale-free characteristics and preserve significant correlations between genes. (**d**) The cluster dendrogram, which displays the clustering of genes, with genes assigned to different modules based on their expression similarities. Different colors at the bottom represent gene modules identified through the dynamic tree cut method, where each color indicates a group of genes that tend to behave similarly. The dendrogram above shows the hierarchical clustering process, with the height on the y-axis reflecting the degree of dissimilarity between genes—taller branches indicate greater differences. The merging of color blocks at the bottom shows how modules are grouped or related to each other based on their gene expression patterns. (**e**) The module-trait relationships heatmap, which displays the correlation between different gene modules and traits. Each module is represented with a correlation coefficient to the traits, indicated by both color and numerical values. Red represents positive correlation, green represents negative correlation, and the deeper the color, the stronger the correlation. R represents the resistant phenotype, S represents the susceptible phenotype, and the healthy group is denoted by NC.

**Figure 3 vetsci-12-00479-f003:**
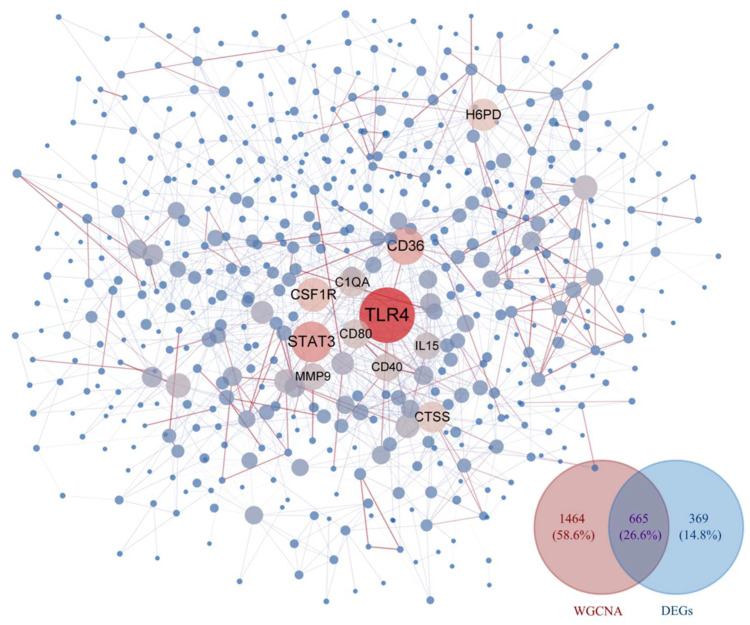
PPI network and Venn diagram. By comprehensively screening genes from the MEbrown and MEblack modules in WGCNA along with DEGs, a protein–protein interaction network was constructed using the 665 genes shared between them. In this network, larger dark red nodes indicate a higher degree of connectivity within the network, while the redder the edges between nodes, the more reliable the interaction between them (based on the combined score from the STRING database). Conversely, smaller nodes suggest poorer connectivity of that node within the network, and the bluer the edges between nodes, the less reliable their interactions are.

**Figure 4 vetsci-12-00479-f004:**
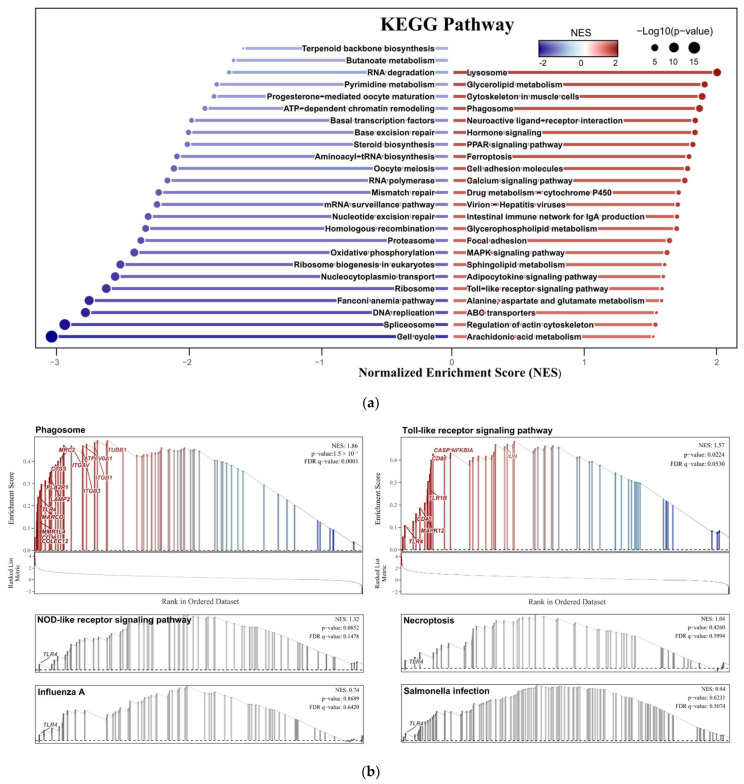
(**a**) Bidirectional column chart. Displays 48 signaling pathways from the GSEA enrichment analysis based on the KEGG database, with those significantly activated (red) and suppressed (blue) identified (|Normalized Enrichment Score, NES| ≥ 1.5, *p* ≤ 0.05); (**b**) Single-gene enrichment analysis result. GSEA results of the six signaling pathways involved in the *TLR4*, of which the gray ones are insignificant pathways (*p* ≥ 0.05).

**Figure 5 vetsci-12-00479-f005:**
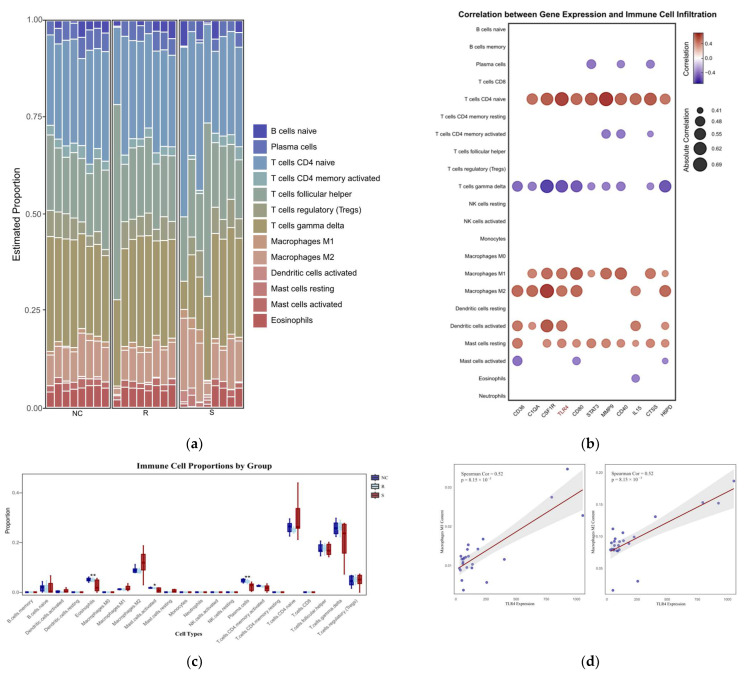
(**a**) The immune cell stacking chart displays the relative abundance of different types of immune cells in each thymus sample. R represents the resistant phenotype, S represents the susceptible phenotype, and the healthy group is denoted by NC. (**b**) The correlation heatmap analyzes the relationships between 11 core genes in the PPI network (Figure 3) and 22 types of immune cells using the Spearman correlation coefficient. The heatmap displays the absolute values of the correlation coefficients (|Cor|), highlighting only the significant correlations between genes and immune cells where |Cor| ≥ 0.4 and *p* ≤ 0.05; (**c**) shows the differences in the proportion of each group of immune cells. R represents the resistant phenotype, S represents the susceptible phenotype, and the healthy group is denoted by NC. * *p* < 0.05 and ** *p* < 0.01 indicate a statistically significant difference from the NC group. (**d**) Scatter plots showing the association of *TLR4* gene expression with M1 and M2 macrophage subsets (calculated using Spearman correlation).

**Figure 6 vetsci-12-00479-f006:**
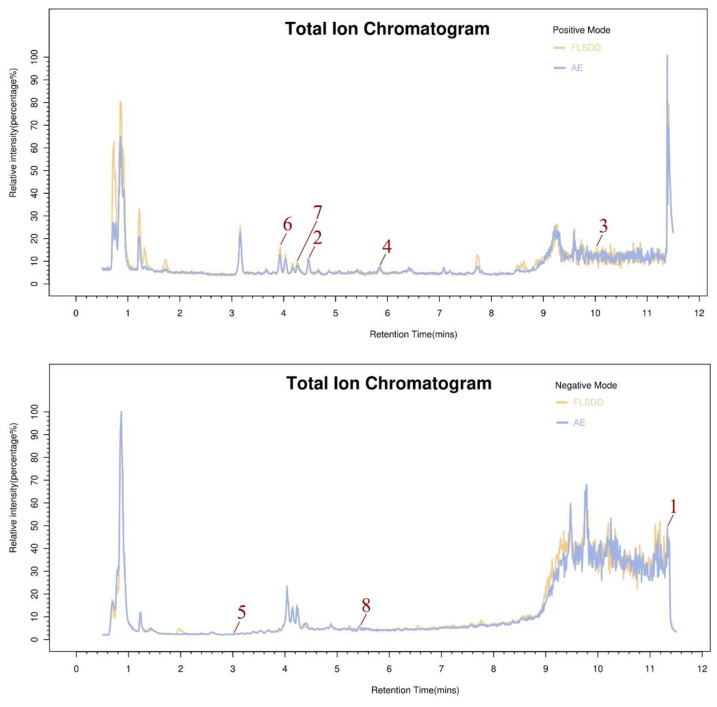
The total ion chromatograms of FLSDD and AE. By matching the results with a database, a total of eight natural compounds targeting TLR4 were identified in LC–MS positive and negative ion modes. As indicated by the numbers in the figure, detailed information of the compounds is shown in Table 3.

**Figure 7 vetsci-12-00479-f007:**
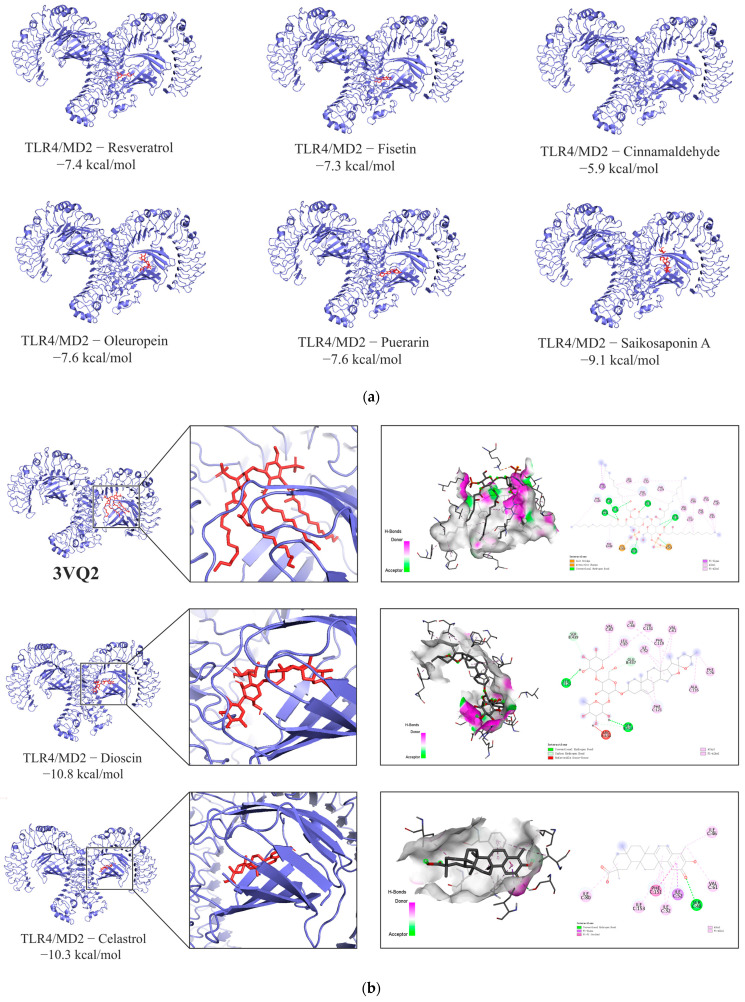
(**a**) The molecular docking results. The blue part is the protein receptor and the red part is the ligand. Show the binding modes of the *TLR4-MD2* complex dimer and six components. (**b**) Binding interactions of LPS and natural compounds with the *TLR4-MD2* complex dimer. The PDB entry *3VQ2* demonstrates the binding interaction between LPS and the *TLR4-MD2* complex dimer. Additionally, the binding modes of two natural compounds (*Dioscin* and *Celastrol*) with the same protein receptor from *3VQ2* are shown alongside it.

**Figure 8 vetsci-12-00479-f008:**
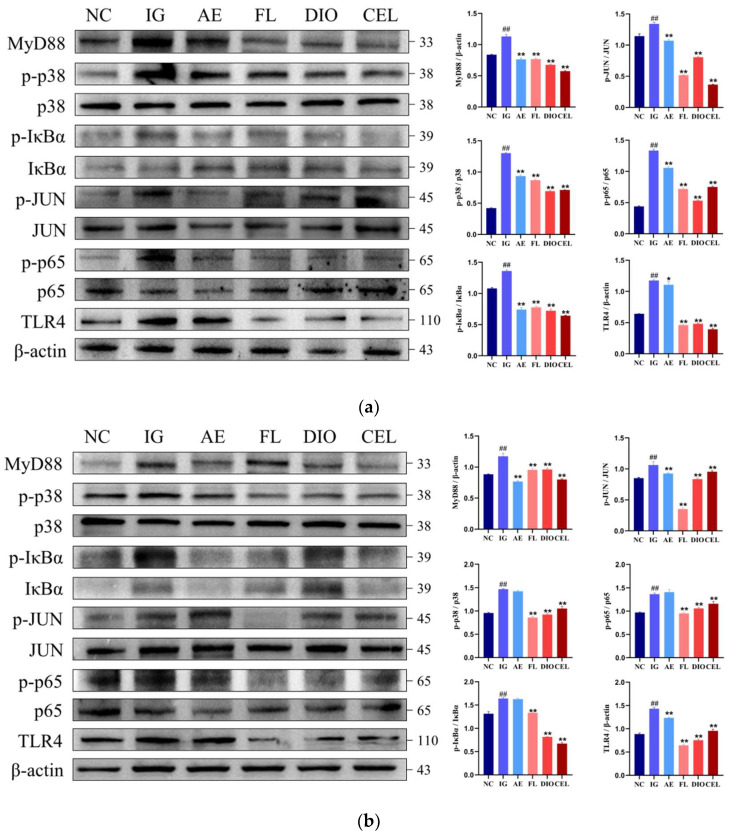
NC denotes the control group, IG denotes the APEC infection group, AE denotes the dregs aqueous extract treatment group, FL denotes the fermented liquid treatment group, and DIO and CEL denote the Dioscin and Celstrol treatment groups, respectively. (**a**) Protein expression levels in chicken macrophages *HD11*. (**b**) Protein expression levels in the chicken duodenum. ## *p* < 0.01 indicate a statistically significant difference from the NC group; * *p* < 0.05 and ** *p* < 0.01 indicate a statistically significant difference from the IG group. WB original figures see Appendix A.

**Figure 9 vetsci-12-00479-f009:**
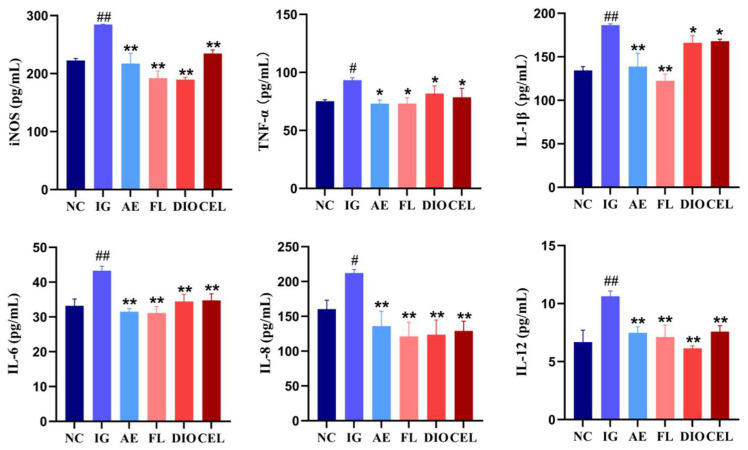
ELISA detection of serum inflammatory factor levels in each treatment group, # *p* < 0.05 and ## *p* < 0.01 indicate a statistically significant difference from the NC group; * *p* < 0.05 and ** *p* < 0.01 indicate a statistically significant difference from the IG group; NC denotes the control group, IG denotes the APEC infection group, AE denotes the dregs aqueous extract treatment group, FL denotes the fermented liquid treatment group, and DIO and CEL denote the Dioscin and Celstrol treatment groups, respectively.

**Table 1 vetsci-12-00479-t001:** Cell administration concentration and challenge dose.

ID	Group	Concentration	Challenge Dose
1	NC	—	—
2	AE	50 mg/mL	1 × 10^5^/well
3	FLSDD	50 mg/mL	1 × 10^5^/well
4	CEL	1 μg/mL	1 × 10^5^/well
5	DIO	150 μg/mL	1 × 10^5^/well
7	MG	—	1 × 10^5^/well

All compounds were dissolved in DMSO, and the drug concentration in the FLSDD group was calculated based on the mass ratio of drug residue to water. In 100 mg/mL of FLSDD, the mass ratio of drug residue to water was 1:10. (m_dregs_:m_water_ = 1:10).

**Table 2 vetsci-12-00479-t002:** Chicken administration dose and challenge dose.

ID	Group	Treat	Dosage	Challenge Dose
1	NC	—	—	—
2	AE	Oral	100 mg/mL, 0.5 mL/day/bird	1 × 10^9^ CFU
3	FLSDD	Oral	100 mg/mL, 0.5 mL/day/bird	1 × 10^9^ CFU
4	CEL	Oral	0.1 mg/kg/day/bird	1 × 10^9^ CFU
5	DIO	Oral	10 mg/kg/day/bird	1 × 10^9^ CFU
7	MG	—	—	1 × 10^9^ CFU

All compounds were dissolved in DMSO, and the drug concentration in the FLSDD group was calculated based on the mass ratio of drug residue to water. In 100 mg/mL of FLSDD, the mass ratio of drug residue to water was 1:10. (m_dregs_:m_water_ = 1:10).

**Table 3 vetsci-12-00479-t003:** List of natural compounds targeting *TLR4* in FLSDD and AE.

ID	Compounds Name	CAS	Structures	References
			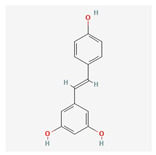	
1	*Resveratrol*	501-36-0		[14,15]

			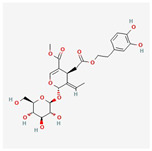	
2	*Oleuropein*	32619-42-4		[16,17]

			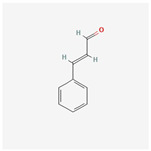	
3	*Cinnamaldehyde*	104-55-2		[18]

			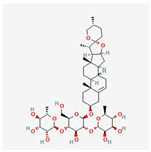	
4	*Dioscin*	19057-60-4		[19,20]

			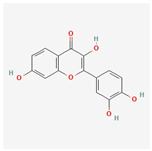	
5	*Fisetin*	528-48-3		[21]

			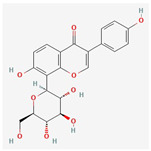	
6	*Puerarin*	3681-99-0		[22]

			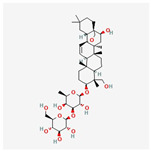	
7	*Saikosaponin A*	20736-09-8		[23]

			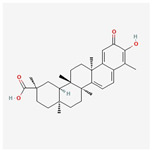	
8	*Celastrol*	34157-83-0		[24]


## Data Availability

The transcriptome dataset GSE69014 can be downloaded from the GEO database (https://www.ncbi.nlm.nih.gov/geo/query/acc.cgi?acc=GSE69014, accessed on 13 May 2025).

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
