# Peer review of "Multi-Omics Unveils Inflammatory Regulation of Fermented Sini Decoction Dregs in Broilers Infected with Avian Pathogenic Escherichia coli"

_vetsci, 2025, doi:10.3390/vetsci12050479_

Round 1
Reviewer 1 Report
Comments and Suggestions for Authors
The manuscript explores a highly complex and relevant approach in addressing one of the greatest challenges of our time: mitigating antimicrobial resistance. Its multidisciplinary perspective is commendable and forward-looking. However, there are several issues the authors need to address before the manuscript can be considered for publication:
- Escherichia coli should be italicized in the title as well as among the keywords.
- In several instances, there is a missing space before in-text reference parentheses throughout the manuscript.
- Line 84: The term “gut flora” is outdated; it is recommended to consistently use “gut microbiome” instead.
- Line 89: The full name of Lactobacillus should be written out, as this constitutes its first mention in the main text (the simple summary and abstract are considered separate sections).
- Line 95: Lactobacillus must be italicized.
- Line 100: The abbreviation “APEC” appears for the first time here and should be spelled out.
- All figures must be interpretable independently; therefore, all abbreviations used in figures and figure legends should be clearly defined.
- The terms in vivo and in vitro should be written in italics.
- The Discussion lacks a reflection on the limitations of natural product-based therapies. The authors should consider addressing the variability and standardization challenges of such products. How can one ensure that each batch contains the same quantity and ratio of active compounds? In practice, a key issue is the difficulty in achieving consistent composition, especially given the complex and potentially synergistic effects of multiple components, which are hard to trace systematically.
- The list of abbreviations should be ordered alphabetically.
Author Response
Comments 1: Escherichia coli should be italicized in the title as well as among the keywords.
Response 1: We searched the full text and italicized “ Escherichia coli ” as requested.
Comments 2: In several instances, there is a missing space before in-text reference parentheses throughout the manuscript.
Response 2: In response to this issue, we have carefully checked and revised the entire article.
Comments 3: Line 84: The term “gut flora” is outdated; it is recommended to consistently use “gut microbiome” instead.
Response 3: We have changed "gut flora" to "gut microbiome"
Comments 4: Line 89: The full name of Lactobacillus should be written out, as this constitutes its first mention in the main text (the simple summary and abstract are considered separate sections).
Response 4: We have modified it as requested and written the full name of Lactobacillus
Comments 5: Line 95: Lactobacillus must be italicized.
Response 5: We searched the full text and italicized “Lactobacillus ” as requested
Comments 6: Line 100: The abbreviation “APEC” appears for the first time here and should be spelled out.
Response 6: We have changed "APEC" to “avian pathogenic Escherichia coli”.
Comments 7: All figures must be interpretable independently; therefore, all abbreviations used in figures and figure legends should be clearly defined.
Response 7: We have added annotations to the abbreviations that appear in the figures (lines 264-266; 310-311; 336; 394-396 and 411-413) to improve readability.
Comments 8: The terms in vivo and in vitro should be written in italics.
Response 8: We searched the full text and revised this
Comments 9: The Discussion lacks a reflection on the limitations of natural product-based therapies. The authors should consider addressing the variability and standardization challenges of such products. How can one ensure that each batch contains the same quantity and ratio of active compounds? In practice, a key issue is the difficulty in achieving consistent composition, especially given the complex and potentially synergistic effects of multiple components, which are hard to trace systematically.
Response 9: This is a very useful and pertinent suggestion, and we agree with the reviewer that it is necessary to add a reflection on the limitations of natural product therapies to the Discussion section. We have therefore revised the content of the Discussion section and added this section at lines 523-543.
Comments 10: The list of abbreviations should be ordered alphabetically.
Response 10: We revised and deleted unnecessary abbreviations from the abbreviation list and arranged the abbreviations in alphabetical order.

Reviewer 2 Report
Comments and Suggestions for Authors
Dear Authors
The original scientific work that I had the opportunity to review is of high quality, it deals with the contemporary and extremely current topic of the emergence of Avian Pathogenic Escherichia coli (APEC), which is creating a serious problem in poultry flocks around the world. In order to reduce the appearance of antimicrobial resistance, the authors tried to comprehensively examine the influence of Fermented Liquid of Sini Decoction Dreg (FLSDD) against APEC as well as its anti-inflammatory effect, using a large number of modern methods. The results that the authors obtained and presented in this study are significant, especially through the ONE HEALTH approach.
I presented all observed shortcomings as comments in the Manuscript in PDF format, which I downloaded from the SuSy platform when accepting the review.
The manuscript is well structured, in accordance with the Instructions for Authors. There is an informative Simple Summary and Abstract. The introduction is optimal, the material and methods are completely described precisely (with minor flaws, in certain places details about the manufacturer of the device or equipment are missing). The results are comprehensive, high-quality and significant, very extensive, shown with very high-quality and illustrative Graphics (Figures), sometimes too descriptively titled. Supplementary files are also provided, as well as Original Images for Blots/Gels.
THE ONLY REASON FOR THE PROPOSAL TO ACCEPT THE MANUSCRIPT WITH MAJOR REVISION is the fact that no Conclusions chapter has been written, which, despite the fact that the Discussion chapter is very voluminous, is unacceptable and must be corrected and added by the author to the existing Manuscript.
I have commented on the list of 48 references cited in this Article in the dedicated "window" on the platform, as well as in the uploaded Reviewer's Report (PDF format).
Kind regards

Author Response
Comments 1: Certain places details about the manufacturer of the device or equipment are missing
Response 1: We have made revisions to address this issue and have included detailed information on equipment or reagents where necessary in the article (lines 176-180; 226-228 and 249-251).
Comments 2: Sometimes too descriptively titled.
Response 2: We have made changes to address this issue by streamlining the titles under the figure annotations in the article and adding additional explanatory text instead of directly describing the content of the figure
Comments 3: THE ONLY REASON FOR THE PROPOSAL TO ACCEPT THE MANUSCRIPT WITH MAJOR REVISION is the fact that no Conclusions chapter has been written
Response 3: We have moved the discussion section on the conclusion to the conclusion section (lines 544-559) and added additional reflective arguments to the discussion section (lines 523-543).
Comments 4: I have commented on the list of 48 references cited in this Article in the dedicated "window" on the platform, as well as in the uploaded Reviewer's Report (PDF format).(In all references (n=48) in the list that follows: the name of the Journal must be written as an abbreviation, in italics. Volume of Journal must be in italic, too.Therefore, check all references and match them with the Instructions for Authors.)
Response 4: We have revised the format of the citations to match the journal's requirements, and I have added additional references (lines 692-701) due to the increased content of the Discussion section.

Round 2
Reviewer 2 Report
Comments and Suggestions for Authors
Dear Authors,
Thank you for accepting all the suggestions and correcting the shortcomings according to them. I have included only 2 small suggestions as suggestions in the Article Manuscript PDF document.
Kind regards
